# Dempster–Shafer Fusion Based on a Deep Boltzmann Machine for Blood Pressure Estimation

**Soojeong Lee and Joon-Hyuk Chang \***

Department of Electronic Engineering, Hanyang University, Seoul 04763, Korea; leesoo86@hanyang.ac.kr
\*   Correspondence: jchang@hanyang.ac.kr; Tel.: +82-2-2220-0357

**Abstract:** We propose a technique using Dempster–Shafer fusion based on a deep Boltzmann machine to classify and estimate systolic blood pressure and diastolic blood pressure categories using oscillometric blood pressure measurements. The deep Boltzmann machine is a state-of-the-art technology in which multiple restricted Boltzmann machines are accumulated. Unlike deep belief networks, each unit in the middle layer of the deep Boltzmann machine obtain information up and down to prevent uncertainty at the inference step. Dempster–Shafer fusion can be incorporated to enable combined independent estimation of the observations, and a confidence increase for a given deep Boltzmann machine estimate can be clearly observed. Our work provides an accurate blood pressure estimate, a blood pressure category with upper and lower bounds, and a solution that can reduce estimation uncertainty. This study is one of the first to use deep Boltzmann machine-based Dempster–Shafer fusion to classify and estimate blood pressure.

**Keywords:** oscillometric blood pressure estimation; deep Boltzman machine; machine learning; Dempster–Shafer fusion

---

## 1. Introduction

Most medical offices, hospitals, and home monitoring devices use an oscillometric technique to measure blood pressure (BP) because maintaining a healthy BP is critical to living a long healthy life. The algorithm of maximum amplitude (MAA) is predominantly utilized to estimate the BP mean by monitoring the cuff pressure in the maximum oscillation. However, the MAA using a fixed ratio is insufficient for estimating the BP since the proportion of these fixed characteristics significantly varies according to the rhythm of the heart, movement artifacts, and cuff size [1,2], which leads to uncertainties of the BP in practice in the oscillometric BP measurements [3–5]. Physiological characteristics have a significant impact on these ratios. Recently, Simjanoska et al. [6] studied a method using electrocardiogram (ECG) signals for BP estimation. A simple cuffless technique based on ECG using machine learning was developed by Matsumura et al. [7].

Meanwhile, neural networks (NNs) have been presented to address the fixed ratio problem of using the MAA technique, which does not employ mathematically complicated models [8]. To estimate BP, the NNs can be more robust methods than the mathematical models in terms of artifacts and noise [8]. An NN with a feature-based technique employing a back-propagation algorithm was thus devised [9]. The Gaussian mixture regression (GMR) and improved GMR were further proposed by Lee et al. to estimate BP [4,5]. In recent years, a deep learning technique has emerged as an outstanding trend in machine learning for signal processing areas. Lee et al. [10] introduced a method based on deep learning [11] to estimate the BP when only a small sample is available. However, this technique has limitations owing to the input BP data consisting of only five BP sample sets per individual volunteer, which acts as an obstacle when using the deep belief network (DBN) technique. The small number of data cannot ensure desirable identification of the DBN to use the complex nonlinear

model. This critical limitation leads to various problems, such as overfitting, since the DBN comprises the complex nonlinear model, including many layers, parameters, and weights. This problem was addressed by increasing the size of the input data by utilizing the parameter bootstrap scheme [12]. Furthermore, a deep Boltzmann machine (DBM) regression model [13] was recently proposed by Lee et al. [12] to resolve the uncertainty. However, the DBN and DBM models can also cause estimate uncertainties in cases in which many random initialized parameters and functions remain.

Motivated by these issues, a DBM-based [12,13] Dempster–Shafer (DS) fusion [12,14] technique is herein presented to classify and estimate the SBP and DBP categories without characteristic ratios. DBM is an advanced technology of the accumulation of restricted Boltzmann machines (RBMs) [13]. Unlike DBN [13], each unit in the middle layer of the DBM acquires signal up and down to prevent uncertainty at the inference step [12]. More recently, the concepts of fusion and ensemble were employed in BP estimation to increase the performance of BP measurements [15–17]. DS fusion enables the evidence uncertainty to have upper and lower bounds that can be incorporated as a method for combined independent estimation of the observations, while also enabling the clear achievement and increased confidence in a given DBM estimate [16]. Consequently, we provide an accurate BP category and a methodology that can reduce the estimation uncertainty. Specifically, this research provides the following contributions and improvements over that presented in [10,12]:

- A methodology to acquire BP estimates utilizing oscillometric BP data is provided, and a BP category using DBM-based DS fusion [14,15] is presented.
- The DS combination is comprised of fused independent estimations of the observations. Moreover, the confidence increases for a given DBM estimate are clearly observed [16] to solve uncertainty to boost the advantage of each DBM classifier and compensate the limitations.
- Our methodology provides upper and lower bounds from the clustered BP measurements using the *k*-medoids algorithm [18] to reduce the BP estimation uncertainty.
- This approach can mitigate the standard deviation of errors (SDEs) of SBP and DBP by 10.4% and 10.1% contrasted to the DBM single estimator [12]. This finding indicates that the DBM-based DS fusion estimator is superior to that of the single DBM estimator in terms of addressing the BP measurement uncertainty.

In Section 2, signal processing and feature data selection from oscillometric BP measurements are expressed. The proposed DBM-based DS fusion method with artificial features is presented in Section 3. Section 4 describes the experimental scenario, measurement methodology, and results. Finally, discussion and conclusions are provided in Section 5.

## 2. Signal Processing and Feature Selection from Oscillometric Wave Signals

This study was confirmed by a research ethical committee of the institution, and every volunteer signed an informed consent rule prior to measurements according to the BP measurement protocol of the institutional research ethical board. The BP data were recorded from 85 subjects between the ages of 12 and 80 years that had no history of cardiovascular disease. Among them, 37 were females and 48 were males. Five sets of BP measurements per subject were collected using a measuring device according to the recommendations of the Sphygmomanometer Committee (SP10) standard [5,19]. Measurements obtained by two nurses were averaged to provide single target SBP and single target DBP values. In this work, the feature data were extracted from the OMW signal and envelope as shown in the third box of Figure 1. First, the features were obtained from the OMW signals and oscillometric envelopes for use as input data to estimate the target SBP and DBP. The target BPs had to be converted into the BP classifier category. A summary of the feature vectors is provided in [12].

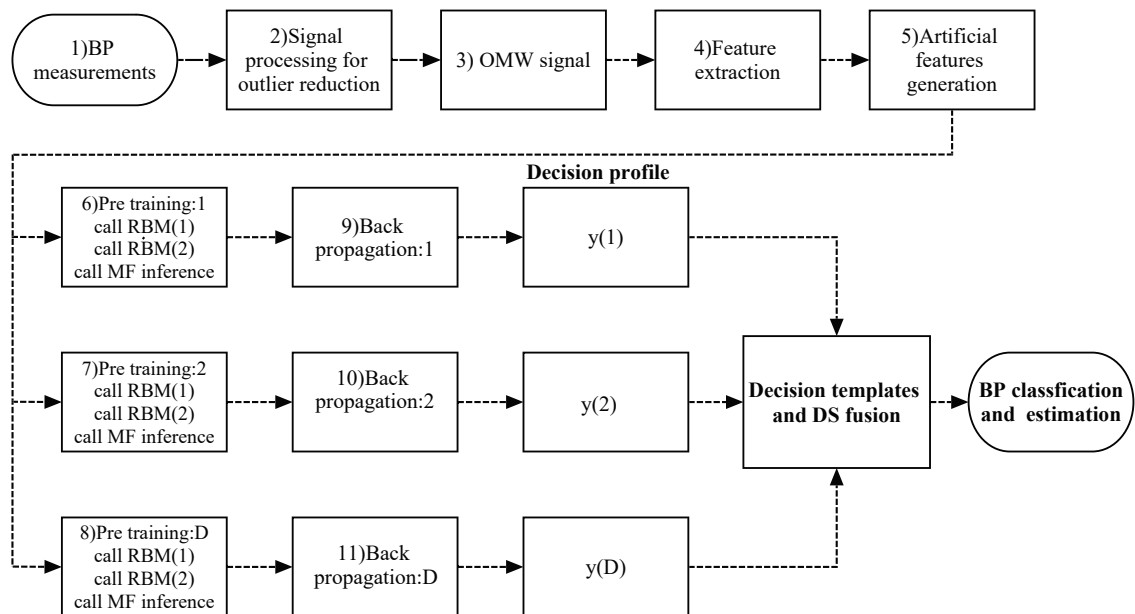

**Figure 1.** Upper boxes: Procedures of BP (blood pressure) measurement, signal processing, oscillometric wave (OMW) signal, feature extraction, and artificial features generation. Right boxes (6 to 11) are DBMs (deep Boltzmann machine). Other boxes: DS (Dempster–Shafer) fusion to decrease uncertainty, where **y**(1) is the output vector of the first classifier, and *D* represents the number of classifiers. DS fusion is further explained in Section 3.

## 3. DBM-Based DS Fusion

### 3.1. Artificial Input Data

Because only five BP measurements per individual participant were recorded, an artificial feature was created by the bootstrap technique presented in [20]. That technique is a recently developed method to make certain kinds of statistical inferences. In this work, we suppose that $\mathbf{S} = [s_1, ..., s_N]$ denotes the sampling distribution of $\mathbb{F}$ with unknown parameters $[\theta, \sigma^2]$. In time, $\mathbb{F}$ is estimated by $\widehat{\mathbb{F}}(\widehat{\theta}, \widehat{\sigma}^2 | \mathbf{S})$, where the mean and variance are represented by $\mathbb{E}(\theta | \mathbf{S}) \simeq \widehat{\theta} = \bar{x} = \frac{1}{N} \sum_{n=1}^{N} s_i$ and $\mathbb{E}(\sigma^2 | \mathbf{S}) \simeq \widehat{\sigma}^2 = [\frac{1}{N-1} \sum_{n=1}^{N} (x_n - \bar{x})^2]$. Here, $\widehat{\mathbb{F}} \simeq \mathcal{N}(\widehat{\theta}, \widehat{\sigma}^2)$ is approximated by a Gaussian distribution, which is a parametric bootstrap [20]. It is employed to create the artificial data $\mathbf{x} = [\theta_1^*, \theta_2^*, ..., \theta_B^*]$ acquired using the real data **x**. Details on the artificial feature created by the bootstrap technique are provided in [3,10,20].

### 3.2. Statistical Analysis

To utilize the cumulative distribution function (CDF) of the artificial feature, the normality of each CDF was clearly investigated. The Kolmogorov–Smirnov (KS) test [21] was thus conducted to evaluate the normality of each CDF. We suppose that $\mathbb{F}^*$, a CDF of an artificial feature vector $[\theta_1^*, \theta_2^*, ..., \theta_B^*]$, is obtained from the sampling distribution, $\mathbb{F}$, where *B* is the number of replications. The KS test result is expressed as the probability of measuring the agreement between the hypothesis and the distribution of the artificial features [5]. The CDFs of the artificial features are confirmed as Gaussian distributions through the test results shown in Figure 2, where the CDF plots almost fit the normal distribution. As B increases, the CDF approaches the normal distribution more closely. The $\mathbb{H}_0$ supposes that the CDF of an artificial feature follows a roughly Gaussian distribution, whereas the $\mathbb{H}_1$ supposes that the distribution of an artificial feature does not approximate a Gaussian distribution [22]. In addition, we verified that the KS test results of all artificial features were zero. Thus, we did not reject the $\mathbb{H}_0$ at

the $\alpha$(=0.05) significance level. We could thus assume that the artificial features had similar statistical properties as their original features.

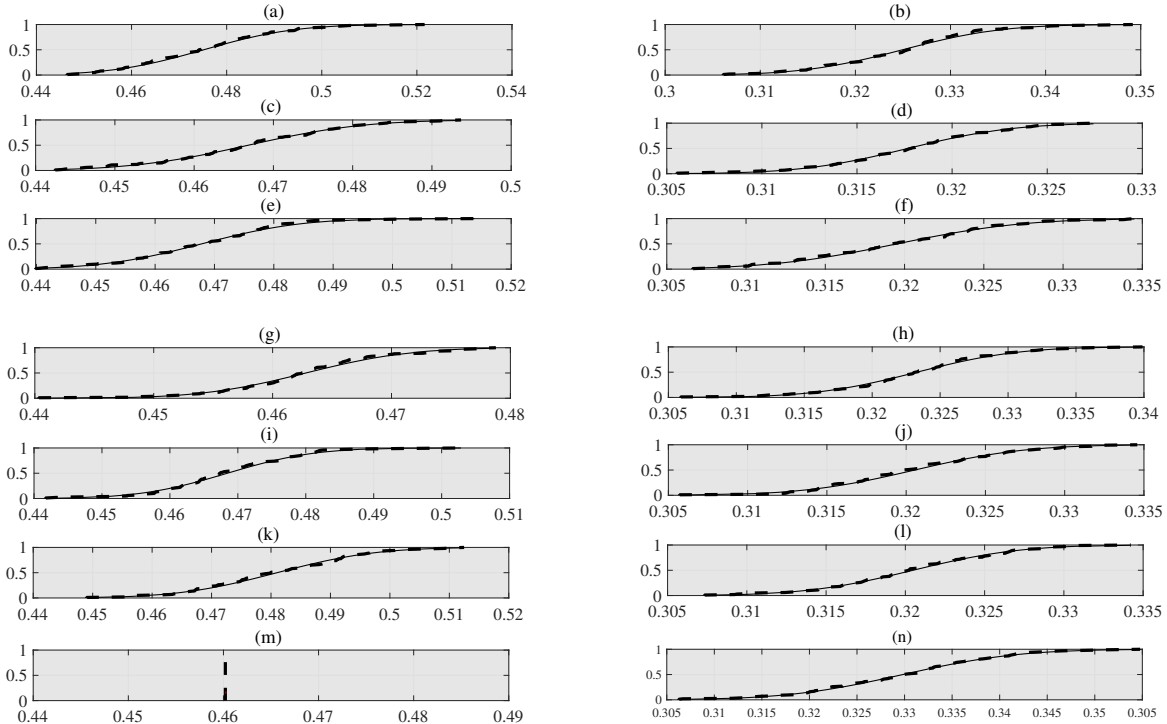

**Figure 2.** The CDF (cumulative distribution function) plots of artificial features (**a**) MAP (mean arterial pressure); (**b**) AR; (**c**) AE; (**d**) LE; (**e**) MA; (**f**) $\sigma$1; (**g**) $\sigma$2; (**h**) LMAP; (**i**) AR2; (**j**) AR3; (**k**) AE2; (**l**) AE3; (**m**) gender; (**n**) HR; and the subject's age are investigated as the features and generated using the bootstrap technique with replication numbers ($B$ = 100) from one volunteer [5,12].

### 3.3. DBM

DBM proposed in [13] was determined to improve the inference during the DBN learning process. As shown in Figure 3, we used a two-hidden-layer DBM to reduce complexity. The energy of the state, $[\mathbf{x}, \mathbf{h}^1, \mathbf{h}^2]$, is defined as:

$$\mathbb{E}(\mathbf{x}, \mathbf{h}^1, \mathbf{h}^2; \theta) = -\mathbf{x}^{\mathrm{T}} W^1 \mathbf{h}^1 - \mathbf{h}^{1\mathrm{T}} W^2 \mathbf{h}^2, \tag{1}$$

where $\theta = [W^1, W^2]$ are parameters that express the symmetric interaction terms of the input units to hidden units, and the hidden units to hidden units [13]. In addition, $i$, $j$, and $m$ denote the indexes of the input unit, hidden unit $\mathbf{h}^1$, and hidden unit $\mathbf{h}^2$, respectively. The probability is assigned to an input data, $\mathbf{x}$, defined by:

$$p(\mathbf{x}; \theta) = \frac{\sum_{\mathbf{h}^1, \mathbf{h}^2} \exp(-\mathbb{E}(\mathbf{x}, \mathbf{h}^1, \mathbf{h}^2; \theta))}{\sum_{\mathbf{x}} \sum_{\mathbf{h}^1, \mathbf{h}^2} \exp(-\mathbb{E}(\mathbf{x}, \mathbf{h}^1, \mathbf{h}^2; \theta))}. \tag{2}$$

The distributions over the input vector and the two sets of hidden units are expressed by:

$$p(h_j^1 = 1 | \mathbf{x}, \mathbf{h}^2) = \phi\left(\sum_i W_{ij}^1 x_i + \sum_m W_{jm}^2 h_j^2\right), \tag{3}$$

$$p(h_m^2 = 1 | \mathbf{h}^1) = \phi\left(\sum_j W_{im}^2 h_i^1\right), \tag{4}$$

$$p(x_i = 1 | \mathbf{h}^1) = \phi\left(\sum_j W_{ij}^1 h_j^1\right), \tag{5}$$

where $\phi(*) = 1/(1 + \exp(-*))$. The generative probability is defined by:

$$p(\mathbf{x}; \theta) = \sum_{\mathbf{h}^1} p(\mathbf{h}^1; \mathbf{W}^1) p(\mathbf{x}|\mathbf{h}^1; \mathbf{W}^1), \tag{6}$$

where $p(\mathbf{h}^1; \mathbf{W}^1) = \sum_{\mathbf{x}} P(\mathbf{x}|\mathbf{h}^1; \mathbf{W}^1)$ denotes an implicit prior probability over $\mathbf{h}^1$ given by the parameters. The second RBM in the stack substitutes $p(\mathbf{h}^1; \mathbf{W}^1)$ by $p(\mathbf{h}^1; \mathbf{W}^2) = \sum_{\mathbf{h}^2} p(\mathbf{h}^1, \mathbf{h}^2; \mathbf{W}^2)$. When the second RBM is correctly initialized, $p(\mathbf{h}^1; \mathbf{W}^2)$ is a better probability of all posterior distributions of $\mathbf{h}^1$, where the all posterior is the non-factorial mixture of the factorial posteriors as $\frac{1}{N} \sum_n p(\mathbf{h}^1|\mathbf{x}_n; \mathbf{W}^1)$. Here, $N$ denotes the size of the input vector. Note that it is possible to infer $P(\mathbf{h}^1; \mathbf{W}^1, \mathbf{W}^2)$ through averaging the two models of $\mathbf{h}^1$, which is approximately performed by utilizing the $1/2\mathbf{W}^1$ bottom-up approach and $1/2\mathbf{W}^2$ top-down approach because the second RBM replaces $p(\mathbf{h}^1; \mathbf{W}^1)$ with a better model [13]. The pre-training is presented [13] to learn a stack of RBMs. The weights of a double input unit to a hidden unit are doubled, as shown in Figure 3. The probabilities of the hidden and input units are defined by

$$p(h_j^1 = 1|\mathbf{x}) = \phi\left(\sum_i W_{ij}^1 x_i + \sum_i W_{ij}^1 x_i\right), \tag{7}$$

$$p(x_i = 1|\mathbf{h}^1) = \phi\left(\sum_j W_{ij}^1 h_j^1\right). \tag{8}$$

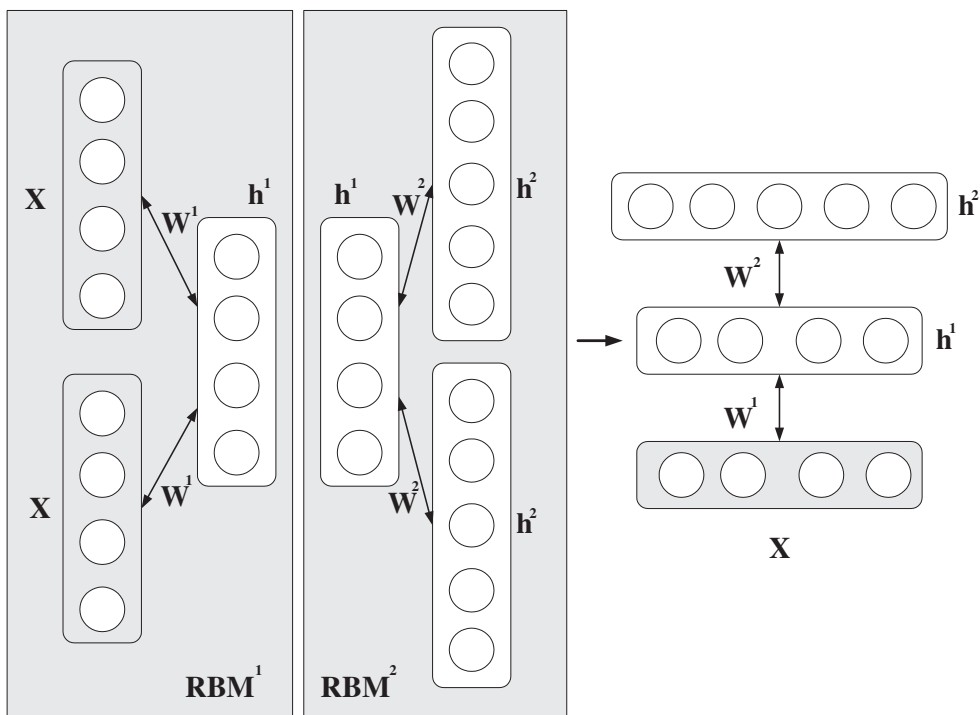

**Figure 3.** DBM (deep Boltzmann machine) is built by using RBMs (restricted Boltzmann machine), where **W** is a weighted parameter [12,13].

The conditional probabilities of this model are given as

$$p(h_j^1 = 1|\mathbf{h}^2) = \phi\left(\sum_m W_{jm}^2 h_m^2 + \sum_m W_{jm}^2 h_m^2\right), \tag{9}$$

where two modules are combined into a single module that leads to the following conditional distribution of $\mathbf{h}^1$:

$$p(h_m^2 = 1|\mathbf{h}^1) = \phi\left(\sum_j W_{jm}^2 h_j^1\right),\tag{10}$$

$$p(h_j^1 = 1|\mathbf{x}, \mathbf{h}^2) = \phi\left(\sum_i W_{ij} x_i + \sum_m W_{jm}^2 h_m^2.\right)\tag{11}$$

The conditional distribution defined by the coupled module is exactly the same one that is given as Equations (3)–(5). Hence, the two coupled RBMs build an undirected model with symmetric weights as a DBM. In particular, each hidden unit is performed in a single with bottom-up fashion, doubling the bottom-up information to compensate for the lack of top-down feedback. This technique is used to initialize average field reasoning and can converge much faster than using arbitrary initialization [13]. Details on the mean field approach is referenced in [13]. Therefore, the RBM finds the best method to assign the initial parameters.

Our goal is to classify the blood pressure category that belongs to the feature vector of the subject when input $\mathbf{x}$ is given. The number of classes to be classified in the DBM output layer is the same as $K$. The output of the $k$th unit of the output layer is given using the activation function of this layer as follows:

$$p(\mathbf{C}_k|\mathbf{x}) = \mathbf{y}_k = \frac{\exp(p(h_k^2 = 1|\mathbf{h}^1))}{\sum_{m=1} \exp(p(h_m^2 = 1|\mathbf{h}^1))}.\tag{12}$$

The above function is called the soft maximum function [18], where $\mathbf{C}_1, ..., \mathbf{C}_K$ denote the classes to be classified. The input $\mathbf{x}$ is then classified as a class showing the highest probability. Assuming that the output is represented by the vector $\Omega_n = [\omega_{n1}, ..., \omega_{nK}]^T$ of the $K$-lists of the binary values, each element is equal to one when the corresponding class is the correct class or versa. Therefore, the posterior distribution is expressed as:

$$p(\Omega|\mathbf{x}) = \prod_{k=1}^K p(\mathbf{C}_k|\mathbf{x})^{\omega_k}.\tag{13}$$

By Equation (13), we can derive the likelihood of $\mathbf{W}$ for the training data:

$$L(\mathbf{W}) = \prod_{n=1}^N \prod_{k=1}^K p(\mathbf{C}_k|\mathbf{x}_n)^{\omega_{nk}}.\tag{14}$$

In these procedures, initial parameter, weights and biases are set for all parameters. Next, stochastic gradient descent (SGD) is conducted as the fine adjustment of the second procedure [23]. After the first procedure, the parameter $\mathbf{W}$ for each layer obtained by two RBMs is utilized as an initial parameter to fine-tune the use of the SGD technique [11]. The objective function uses the cross-entropy criterion by employing a mini-batch scaled SGD between the estimated BP and reference BP defined as:

$$\mathbb{E}(\mathbf{W}) = -\sum_{n=1}^N \sum_{k=1}^K \omega_{nk} \log(p(\mathbf{C}_k|\mathbf{x})|\mathbf{W}),\tag{15}$$

where $N$ represents the input data size, and $\mathbf{W}$ denotes the weights and bias parameters to be learned at each layer.

### 3.4. DS Fusion

In this paper, we propose a fusion method of three DBM classifiers based on DS fusion theory. We offer a technique to handle uncertainty in the observation fusion to enhance the advantage of each classifier and compensate the weakness. The DBM classifier $\mathbf{f}_n$ based on DS fusion is used to decrease

the observation uncertainty through a combination of decision rules. Here, we omit the details of DS theory [14,15,24,25].

Suppose that we have an input data, $\mathbf{x}$, and a different classier $\mathbf{f}_d, = 1, ..., D$. In addition, we suppose that each DBM classifier creates an output $\mathbf{y}_d \in \mathbf{R}_k, \mathbf{y}_d = \mathbf{f}(\mathbf{x})_d$, where $K$ denotes the number of classes. We assume that, for each DBM classifier $\mathbf{f}_d$ and each candidate class $k$, we compute the value $b_k(\mathbf{y}_d) = b_k(\mathbf{f}_d(x))$, which is some measure of *belief* for the proposition ($\mathbf{y}_d$ is of class $k$). As mentioned before, we find the posterior probability of each class to calculate *belief* given the output data. We must build a multidimensional distribution for the output vector to estimate the conditional probability distribution for all $K$ classes. Assume that $\mathbf{x}_k$ is a subset of training data equivalent to class $k$, and $\bar{\mathbf{y}}_{d,k}$ is the mean vector for a set $\mathbf{f}_d(x_k)$ for each $\mathbf{f}_d$ with each class $k$. Thus, $\bar{\mathbf{y}}_{d,k}$ is a reference vector for each class. Now, we compute the "proximity" $\Phi$ between $\bar{\mathbf{y}}_{d,k}$ and $y_d$ as given by:

$$\Phi(\bar{\mathbf{y}}_{d,k}, \mathbf{y}_d) = d_{d,k} = \frac{(1 + \| \bar{\mathbf{y}}_{d,k} - \mathbf{y}_d \|^2)^{-1}}{\sum_{k=1}^{K}(1 + \| \bar{\mathbf{y}}_{d,k} - \mathbf{y}_d) \|^2)^{-1}}. \tag{16}$$

Actually, $\Phi(\bar{\mathbf{y}}_{d,k}, \mathbf{y}_d)$ is the maximum value that varies between 1 and 0 when the output vector matches the reference vector. Thus, $\Phi(\bar{\mathbf{y}}_{d,k}, \mathbf{y}_d)$ must be converted *belief* $b_k(\mathbf{y}_d)$. We consider a frame of discernment $\Omega = \{\omega_1, ..., \omega_K\}$, where $\omega_k$ denotes the hypothesis that ($y_d$ is of class $k$). Moreover, $d_{d,k}$ can express *belief pro*-hypothesis $\omega_k$, and all $d_{d,i}$, and $i \neq k$ can express *belief pro*$\neg\omega_k$ or *contra* $\omega_k$ for any DBM classifier $\mathbf{f}_d$ with each class $k$. Here, we use $d_{d,k}$ as a degree of support with $\omega_k$. A basic probability assignment (BPA) is thus defined as

$$\mathbf{m}_k(\omega_k) = d_{d,k}, \quad \mathbf{m}_k(\Omega_k) = 1 - d_{d,k} \tag{17}$$

where $d_{d,i}$ is also a degree of support with $\neg\omega_k(i \neq k)$. The fusion of these support functions with $\neg\omega_k(i \neq k)$ denotes a separable support function with the degree of support $(1 - \prod_{i \neq k}(1 - d_{d,k}))$. The corresponding BPA is given by:

$$\mathbf{m}_{\neg k}(\neg\omega_k) = (1 - \prod_{i \neq k}(1 - d_{d,k})), \tag{18}$$

$$\mathbf{m}_{\neg k}(\Omega_k) = 1 - \mathbf{m}_{\neg k}(\neg\omega_k). = \prod_{i \neq k}(1 - d_{d,k}). \tag{19}$$

To fuse our knowledge with respect to $\omega_k$, we acquire the *belief* $\mathbf{m}_k \oplus \mathbf{m}_{\neg k}$, *pro* $\omega_k$, where $\oplus$ is an orthogonal sum given by:

$$b_k(\mathbf{y}_d) = \frac{d_{d,k}(x) \prod_{i \neq k}(1 - d_{d,i})}{1 - d_{d,k}[1 - \prod_{i \neq k}(1 - d_{d,i}))]}. \tag{20}$$

The *belief* degree is acquired for each classifier. It can be combined by the DS theory of fusion and should be an orthogonal sum to obtain the final degrees of support, which are:

$$b_k(\mathbf{x}) = b_k(\mathbf{y}_1) \oplus ... \oplus b_k(\mathbf{y}_D). \tag{21}$$

Therefore, we can assign class $j$ to the input vector $\mathbf{x}$ if $b_j = \max_{1,...,K} b_k(\mathbf{x})$.

## 4. Experimental Results and Statistical Analysis

To implement this work, the 15 feature vectors acquired from the OMWs and OMW envelopes [12] were used as summarized in [12]. We selected the number of BP categories as $K(=10)$ to build a sophisticated classifier for SBP and DBP, respectively, as shown in Table 1. To classify SBP and DBP target values determined by the nurses, we used the *k*-medoids algorithm in Matlab® 2017. The *k*-medoids method imposed a clustering structure on the SBP and DBP target values.Therefore, we obtained the *k* cluster medoid locations as shown in the second and third columns of Table 1.

The algorithm is renowned as as being real-valued and less sensitive to the presence of noise compared to *k*-means [18]. We conducted training and testing experiments to evaluate DBM-based DS fusion technology. The subjects' BP measurements were sequentially divided into training data (340 sets from 68 subjects) and test sets (85 sets from 17 subjects). Indeed, the five measurements obtained from the individual subjects is a very a small number for the input data in the deep learning. Thus, we adopted the artificial data acquired using the real data. We used the unseen real data to evaluate the new technique in the test step.

**Table 1.** BP (blood pressure) categories are represented by the *k*-medoids algorithm corresponding to cluster locations obtained by nurse BP measurements, where L and U denote lower and upper bounds.

| BP (mmHg) | SBP | DBP | SBP L | SBP U | DBP L | DBP U |
|---|---|---|---|---|---|---|
| category 1 | 81.6 | 45.5 | 79.3 | 92.4 | 42.9 | 49.1 |
| category 2 | 92.8 | 50.5 | 93.0 | 96.4 | 50.2 | 55.7 |
| category 3 | 98.9 | 56.6 | 96.5 | 101.0 | 55.8 | 60.2 |
| category 4 | 103.7 | 60.3 | 101.1 | 105.1 | 60.2 | 63.4 |
| category 5 | 107.9 | 64.1 | 105.2 | 109.6 | 63.5 | 66.7 |
| category 6 | 112 | 68.2 | 109.8 | 115.5 | 66.8 | 69.9 |
| category 7 | 117.7 | 71.0 | 115.6 | 119.6 | 70.0 | 75.3 |
| category 8 | 123.9 | 77.0 | 119.8 | 126.4 | 75.4 | 80.5 |
| category 9 | 130.7 | 81.6 | 126.8 | 134.5 | 80.6 | 85.7 |
| category 10 | 139.8 | 98.2 | 135.5 | 144.8 | 88.3 | 98.6 |

We increased the epoch number from 10 to 100. As a result of the experiment, the mean square error (MSE) was rapidly decreased until 20 epochs. From that point, it was slightly increased and decreased until 100 epochs. We then tuned the performance of the proposed method on the identical conditions with a different number of hidden units from 16 up to 128 owing to the dimension $\mathbb{R}(=15)$ of the input features. The best benefit of the proposed method was found at 64 hidden units, which showed that a small number of hidden units could cause a higher mean absolute error (MAE) and standard deviation of error on account of underfitting, whereas a large number of hidden units could lead to an increased MAE and SDE because of overfitting. Next, we chose the number of hidden units (=64) and number of epochs (=100) to compare the running time between the DBM-based DS fusion and DBM techniques. We summarized the input order of the new technique as 15, the output order as 20, and the hidden unit as 2. The total of the training and test times was calculated based on Matlab® 2017 [26]. It was adequately short for real BP application because it processed 8500 pieces of data in 6130.8 s. From this result, the DBM-based DS fusion called for a higher complexity time than the DBM technique, which was owing to the fusion process.

For comparison, we first investigated the accuracy of the BP classifiers using a support vector machine (SVM), SVM with DS fusion (SVMDS), DBN, DBN with DS fusion (DBNDS), DBM, and proposed DBM-based DS fusion (DBMDS), as presented in Table 2. We found that BP classification probability (sensitivity) versus false alarm probability (1-specificity) across varying cut-off produces a receiver operating characteristic (ROC) curve in the square as shown in Figure 4. The characteristic of the ROC curve gradually shows better performance nearer to the upper left. Therefore, the fusion SBP and DBP estimators have a better performance than the single SBP and DBP estimators. The new fusion scheme for SBP and DBP measurements was evaluated by the mean error (ME) and SDE between the estimated BP categories obtained from the classifier and target BP categories in accordance with the recommendations of the American National Standards Institute (ANSI)/American Association of Medical Instruments (AAMI) standard protocol [19], as shown in Table 3. BP devices are evaluated as passing on a standard protocol [19] basis if the ME is less than 5 mm Hg and the SDE is less than 8 mm Hg versus the reference. Additionally, the British Hypertension Society (BHS) protocol determines the achievement of BP estimates with a cumulative percentage of readings belonging to the mean absolute errors of 5 mmHg, 10 mmHg and 15 mmHg in stethoscope measurements of two trained

observers [27]. Moreover, it classifies the BP measuring device into four classes from A to D, as shown in Table 4.

**Table 2.** Performance comparison of SBP (systolic blood pressure) and DBP (diastolic blood pressure) with respect to classification rate among SVM (support vector machine) [28], SVM with DS (Dempster–Shafer) fusion (SVMDS), DBN (deep belief network) [10], DBN with DS(DBNDS), DBM (deep Boltzmann machine) [12], and the proposed DBM-based DS fusion (DBMDS) scheme.

| Accuracy | SVM | | SVMDS | | DBN | | DBNDS | | DBM | | DBMDS | |
|---|---|---|---|---|---|---|---|---|---|---|---|---|
| **test** | **SBP** | **DBP** | **SBP** | **DBP** | **SBP** | **DBP** | **SBP** | **DBP** | **SBP** | **DBP** | **SBP** | **DBP** |
| test 1 | 0.64 | 0.67 | 0.67 | 0.70 | 0.67 | 0.69 | 0.71 | 0.74 | 0.67 | 0.70 | 0.73 | 0.76 |
| test 2 | 0.66 | 0.68 | 0.68 | 0.71 | 0.68 | 0.72 | 0.70 | 0.75 | 0.69 | 0.71 | 0.73 | 0.76 |
| test 3 | 0.65 | 0.70 | 0.67 | 0.72 | 0.66 | 0.69 | 0.72 | 0.73 | 0.68 | 0.71 | 0.74 | 0.77 |
| test 4 | 0.66 | 0.69 | 0.68 | 0.71 | 0.65 | 0.72 | 0.71 | 0.73 | 0.68 | 0.74 | 0.73 | 0.76 |
| test 5 | 0.65 | 0.69 | 0.68 | 0.71 | 0.69 | 0.73 | 0.71 | 0.74 | 0.72 | 0.74 | 0.74 | 0.77 |
| avg | 0.65 | 0.69 | 0.68 | 0.71 | 0.67 | 0.71 | 0.71 | 0.74 | 0.69 | 0.72 | 0.73 | 0.76 |
| std | 0.01 | 0.01 | 0.01 | 0.01 | 0.02 | 0.02 | 0.01 | 0.01 | 0.02 | 0.02 | 0.01 | 0.01 |

**Table 3.** The results of ME (mean error) and SDE (standard deviation of error) relative to the target results obtained with the conventional MAA (maximum amplitude algorithm), SVM [28], SVMDS, DBN [10], DBNDS, DBM [12], and DBMDS scheme, where the results are the mean values for our test data.

| mmHg | MAA | | SVM | | SVMDS | | DBN | | DBNDS | | DBM | | DBMDS | |
|---|---|---|---|---|---|---|---|---|---|---|---|---|---|---|
| **Test** | **SBP** | **DBP** | **SBP** | **DBP** | **SBP** | **DBP** | **SBP** | **SBP** | **DBP** | **SBP** | **DBP** | **DBP** | **SBP** | **DBP** |
| ME | 0.60 | 1.73 | −0.17 | 0.19 | 0.40 | −0.25 | 0.11 | −0.03 | −0.24 | 0.09 | 0.21 | −0.18 | 0.17 | −0.11 |
| SDE | 9.32 | 7.40 | 7.14 | 5.25 | 6.00 | 5.14 | 6.28 | 5.20 | 5.66 | 4.71 | 5.96 | 4.85 | 5.34 | 4.36 |

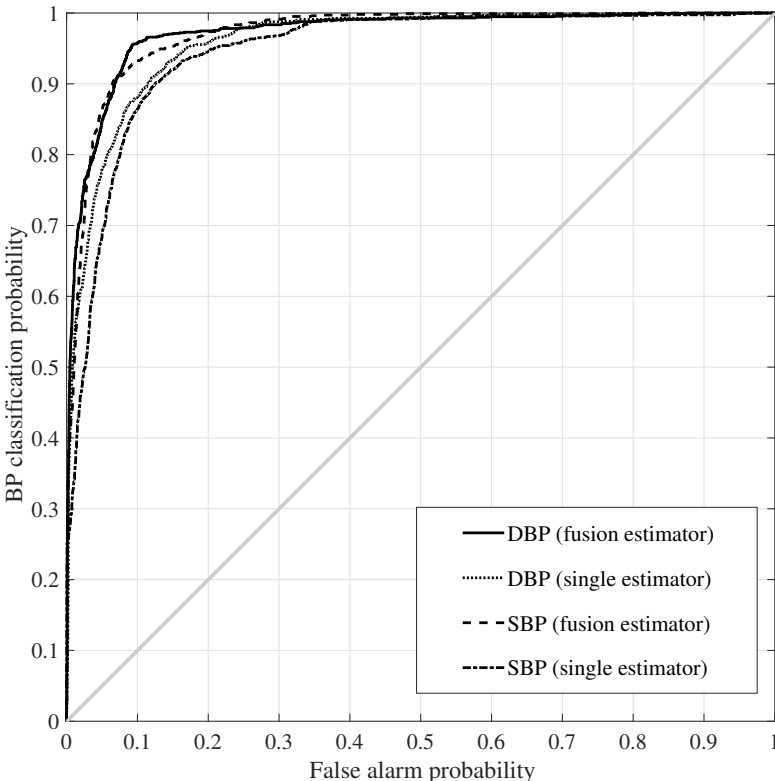

**Figure 4.** ROC (receiver operating characteristic) curve based on experimental conditions using the DBMDS (DBM based DS fusion) estimator vs. DBM single estimator for SBP (systolic blood pressure) and DBP (diastolic blood pressure) classification.

**Table 4.** Grading of the fusion scheme according to the BHS (Brithsh hypertension society) [27] standard employing the results of MAA, SVM [28], SVMDS, DBN, DBNDS, DBM, and DBM based DS fusion schemes on $(5 \times 85 = 425)$ measurements.

| Tests | SBP Absolute Difference (%) | | | DBP Absolute Difference (%) | | | Standard (SBP/DBP) BHS Grade |
|---|---|---|---|---|---|---|---|
| | ≤5 mmHg | ≤10 mmHg | ≤15 mmHg | ≤5 mmHg | ≤10 mmHg | ≤15 mmHg | |
| MAA | 47.86 | 76.90 | 92.38 | 52.38 | 82.86 | 94.05 | C/B |
| SVM | 52.29 | 90.71 | 97.14 | 71.43 | 93.57 | 98.10 | B/A |
| SVMDS | 56.24 | 90.80 | 97.41 | 72.00 | 94.12 | 98.59 | A/A |
| DBN | 62.35 | 88.94 | 97.18 | 72.00 | 94.12 | 98.59 | A/A |
| DBNDS | 64.98 | 91.06 | 97.88 | 74.12 | 94.82 | 99.06 | A/A |
| DBM | 64.52 | 90.82 | 97.18 | 72.24 | 94.38 | 98.59 | A/A |
| DBMDS | 66.12 | 92.47 | 98.82 | 77.20 | 97.19 | 99.81 | A/A |

## 5. Discussion and Conclusions

Under the experimental conditions, we first evaluated the accuracy between the new DBM-based DS fusion and the SBP and DBP target categories obtained from the auscultatory method, as represented in Figure 5 and Table 2. From the results, the DBM-based DS fusion scheme outperformed the SVM, SVMDS, DBN, DBNDS, and DBM single estimator. This implied that the proposed scheme represented 6% in the SBP and 4% in the DBP improvement effect compared to the DBM single estimator. Furthermore, it demonstrated 8% in the SBP and 7% in the DBP improvement effect compared with the SVM technique [28].

In addition, we confirmed that the proposed DBMDS fusion scheme provided more reliable results than the DBM single technique, as shown in Figure 5. This result showed that the estimated uncertainty was effectively reduced, as demonstrated by the solid lines in Figure 5. The ME results for the SBP and DBP computed using the DBM-based DS fusion algorithm were compared with those when using the MAA, SVM [28], SVMDS, DBN, DBNDS, and single DBM technique, as shown in Table 3. The SDEs calculated by the DBM-based DS fusion technique were acquired to be 5.34 and 4.36 mmHg for the SBP and DBP. The SDEs of the new fusion technique were improved by 3.97 and 3.04 mmHg for the SBP and DBP contrasted to those of the MAA, as represented in Table 3. These results expressed that the new fusion technique was superior to that of the single DBM technique with respect to the BP measurement uncertainty. Moreover, we performed an analysis using Bland–Altman plots to evaluate the achievement of the new fusion technique to the stethoscope measurements, as shown in Figure 6. The results in the figure indicate that the estimated BP values obtained by the DBM-based DS fusion technology closely matches the reference SBP and DBP. The range of agreement we use (bold horizontal lines in Figure 6) is for two plots (ME $\pm 2 \times$ SDE) and most of the data is in the range (ME $\pm 2 \times$ SDE). Thus, the achievement imply that the new fusion technique offer precise SBP and DBP contrasted to the single DBM technique. On the basis of above results, the new fusion scheme provided grades of A and A for the SBP and DBP. The achievements of the DBMDS fusion technique were 66.12% (≤5 mmHg), 92.47% (≤10 mmHg), and 98.82% (≤15 mmHg) for the SBP in the given test scenario, and 77.20% (≤5 mmHg), 97.19% (≤10 mmHg), and 99.81% (≤15 mmHg) for the DBP in the test scenario, as represented in Table 4. Therefore, we proved that the probabilities of the BHS criteria were superior to those of the other techniques that obtained grades for the SBP and DBP, as shown in Table 4.

In conclusion, the proposed DBMDS fusion scheme acquired lower SDEs of MEs for the SBP and DBP contrasted with the DBM single estimator. This work thus provides an accurate BP category and offers a solution that can reduce the estimation uncertainty. Although the accuracy and stability are enhanced owing to the proposed DBMDS fusion scheme, our future study will try how the proposed fusion method behaves on individual algorithms.

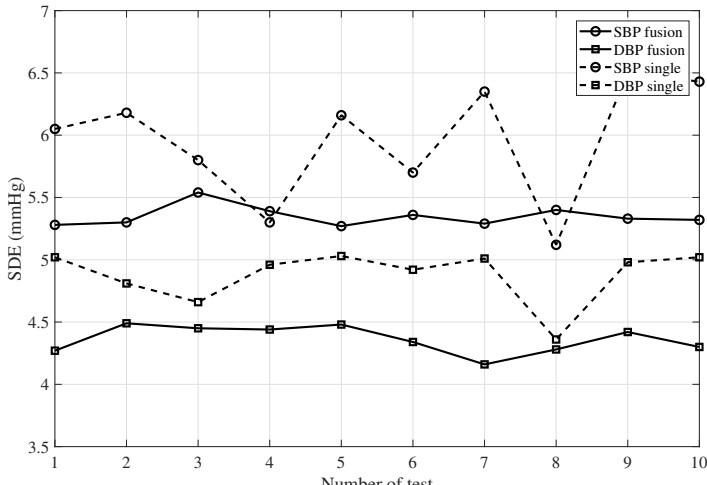

**Figure 5.** Summary of the SDE (ME) obtained utilizing the DBMDS estimator vs. DBM single estimator as the number of test increases according to the AAMI (American association of medical Instruments) standard protocol, where mean (5.34) and standard deviation (STD) (0.08) of SDE obtained from the DBMDS estimator vs. mean (5.96) and STD (0.47) of SDE obtained from the DBM estimator for the SBP, and mean (4.36) and STD (0.11) of SDE obtained from the DBMDS estimator vs. mean (4.85) and STD (0.22) of SDE obtained from the DBM estimator for the DBP.

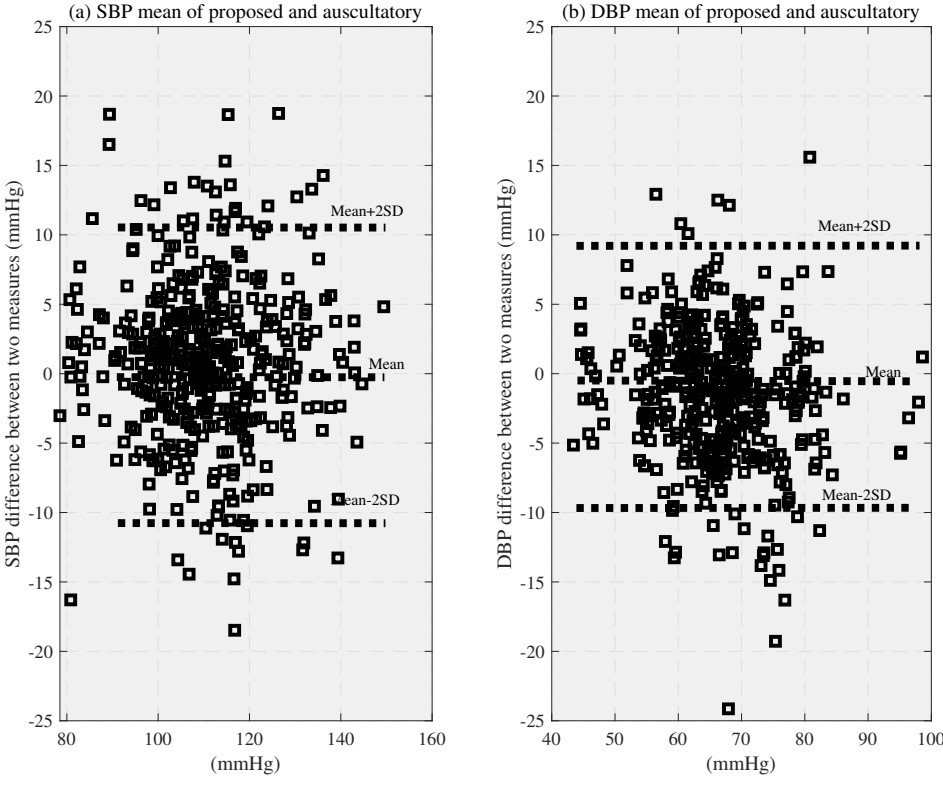

**Figure 6.** Performance comparison between new fusion technique and stethoscope nurse measurements using the Bland–Altman method [5,12,19]. (**a**) Bland–Altman plot for the SBP; (**b**) Bland–Altman plot for the DBP.

**Author Contributions:** Conceptualization, S.L.; methodology, S.L.; software, S.L.; writing—original draft preparation, S.L.; writing—review and editing, J.-H.C.; funding acquisition, J.-H.C.

**Acknowledgments:** This work was supported by the National Research Foundation of Korea(NRF) grant funded by the Korea government(MSIP) (No. 2017R1A2A1A17069651).

**Conflicts of Interest:** The authors declare no conflict of interest.

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
