# Peer review of "Dempster–Shafer Fusion Based on a Deep Boltzmann Machine for Blood Pressure Estimation"

_applsci, doi:10.3390/app9010096_

Round 1

Reviewer 1 Report

This paper proposes Dempster-Shafer (DS) fusion based on a deep Boltzman
machine (DBM) to classify and estimate systolic blood pressure (BP) and diastolic BP categories
using oscillometric blood pressure measurements. The research is interesting, however I have the following concerns whioch are listed below:

1) Introduction needs to be further improved with most recent literature review

2) Fusion of the algorithms are fine however, it will be interesting to see how the proposed method behaves on individual algorithms

3) Please include ROC analysis supported by sensitivity and specificity analysis

4) Please compare the proposed method with other method available in the literature

5) Discussion and conclusion section need to be further improved.

Author Response

Response to the first reviewers’ comments

General

We appreciate very much the valuable comments and suggestions of the reviewers on our paper. In our resubmitted manuscript, we have incorporated all the comments and suggestions made by the reviewers, and have given additional explanations. Our detailed responses are as follows.

1. According to the comment “Introduction needs to be further improved with most recent literature review.”

Answer 1. We included the appropriate literature review such that

Recently, Simjanoska et al. [6] studied a method using electrocardiogram (ECG) signals for BP estimation. A simple cuff less technique based on ECG using machine learning was developed by Matsumura et al. [7].” (lines 23-25, page 2)

2. According to the comment “Fusion of the algorithms are fine however, it will be interesting to see how the proposed method behaves on individual algorithms.”

Answer 2. We also fully agreed about the comment and included a future possibility in Conclusion such as “Although the accuracy and stability are enhanced owing to the proposed DBMDS fusion scheme, our future study will try how the proposed fusion method behaves on individual algorithms.” (lines 209-211, page 12)

3. According to the comments “Please include ROC analysis supported by sensitivity and specificity analysis.”

Answer 3. We included the detailed explanation such that “We found that BP classification probability (sensitivity) versus false alarm probability (1-specificity) across varying cut-off produces a receiver operating characteristic (ROC) curve in the square as shown in Fig. 4. The characteristic of the ROC curve gradually shows better performance nearer to the upper left.

Therefore, the fusion SBP and DBP estimators have a better performance than the single SBP and DBP estimators.” (lines 161-163, page 10) 

4. According to the comments “Please compare the proposed method with other method available in the literature.”

Answer 4. We also compared the proposed fusion technique to most recent literatures such that

[10] S. Lee, and J.-H. Chang, Oscillometric Blood Pressure Estimation Based on Deep Learning, IEEE Transactions on Industrial Informatics, vol. 13, no. 2, pp. 461-472, Apr. 2017.

[13] S. Lee, and J.-H. Chang, Deep Boltzmann Regression with Mimic Features for Oscillometric Blood Pressure Estimation, IEEE Sensors Journal, vol. 17, no. 18, pp. 5982-5993, Set. 2017.

5. According to the comments “Discussion and conclusion section need to be further improved.”

Answer 5. We added a future possibility in Conclusion such as “Although the accuracy and stability are enhanced owing to the proposed DBMDS fusion scheme, our future study will try how the proposed fusion method behaves on individual algorithms.” (lines 209-211, page 12)

Reviewer 2 Report

nice piece of work, you need to make the following modifications foa a better presentation rresult:

-avoid abbreviations in abstract even if you explain them, move them in Introduction section.

-you definitely need to re-arrange the reference list numbering and present them inside the text in a serial order, as it stands now the reader gets confused for example look in line 35-37, line 43 (back to reference [3]) and it continuous like this in lines 79-84 and so on.

-Try to split figure 2 into 2 figures and make plots bigger, you loose information in the way you present them now.

Author Response

Response to the second reviewers’ comments

General

We appreciate very much the valuable comments and suggestions of the reviewers on our paper. In our resubmitted manuscript, we have incorporated all the comments and suggestions made by the reviewers, and have given additional explanations. Our detailed responses are as follows.

1. According to the comments “avoid abbreviations in abstract even if you explain them, move them in Introduction section.”

Answer 1. We removed abbreviations in abstract. (page 1)

2. According to the comments “you definitely need to re-arrange the reference list numbering and present them inside the text in a serial order, as it stands now the reader gets confused for example look in line 35-37, line 43 (back to reference [3]) and it continuous like this in lines 79-84 and so on.”

Answer 2. We checked the reference list numbering.

3. According to the comments “Try to split figure 2 into 2 figures and make plots bigger, you loose information in the way you present them now.

Answer 3. We completely redrawn the figure as shown in Figure 2. (page 4). 

Reviewer 3 Report

The paper is very interesting and describes the development and application of a novel algorithm for BP measurement compared to the gold standard reference.

First of all it is not clear the sentence at the beginning of paragraph 2: do you mean the ethical committee of the institution? if so pleas provide a more clear statement and the study number.

It is not clear if the protocol reported in the paper corresponds to the protocol approved by the committee.

The protocol takes as reference the nurse measurements (5) and describes a method to extract a "bootstrap feature": please clarify this point by better describing what is a bootstrap feature in you study. It seems to be an average OMW that author describes as "artificial".

The protocol does not include any oscillometric device, better a couple of them (one approved and one not approve by ESH, see at http://www.dableducational.org/index.html) so that a comparison can be done to verify the performance of the proposed algorithm with respect to these devices implementing the oscillometric method, that is the goal of the study and of the paper itself.

Data from patient are not clustered: for instance Sex, BMI, Age (this last factor influence arterial stiffness and therefore the BP results) should be analyzed as factors affecting the outcomes.

No information about the computational time is provided: is the proposed solution to be ported into actual devices? if not why and what technological requirements should be improved? 

Author Response

General

We appreciate very much the valuable comments and suggestions of the reviewers on our paper. In our resubmitted manuscript, we have incorporated all the comments and suggestions made by the reviewers, and have given additional explanations. Our detailed responses are as follows.

1. According to the comment “First of all it is not clear the sentence at the beginning of paragraph 2: do you mean the ethical committee of the institution? if so pleas provide a more clear statement and the study number.”

Answer 1. We represented the about it such that “This study was confirmed by a research ethical committee of the institution, and every volunteer signed an informed consent rule prior to measurements according to the BP measurement protocol of the institutional research ethical board.” (lines 70-72, page 3) 

2. According to the comment “It is not clear if the protocol reported in the paper corresponds to the protocol approved by the committee.”

Answer 2. We represented the about it such that “This study was confirmed by a research ethical committee of the institution, and every volunteer signed an informed consent rule prior to measurements according to the BP measurement protocol of the institutional research ethical board.”  (lines 70-72, page 3) 

3. According to the comments “The protocol takes as reference the nurse measurements (5) and describes a method to extract a "bootstrap feature": please clarify this point by better describing what is a bootstrap feature in you study. It seems to be an average OMW that author describes as "artificial".”

Answer 3. Dear reviewer! We didn’t use “bootstrap feature” in the manuscript. We used artificial features created by the parametric bootstrap technique such that “Because only five BP measurements per individual participant were recorded, an artificial feature was created by the bootstrap technique presented in [21].” (lines 83-84, page 4), If you find “bootstrap feature”, please let me know.

4. According to the comment “The protocol does not include any oscillometric device, better a couple of them (one approved and one not approve by ESH, see at http://www.dableducational.org/index.html) so that a comparison can be done to verify the performance of the proposed algorithm with respect to these devices implementing the oscillometric method, that is the goal of the study and of the paper itself.”

Answer 4. We added description with respect to “The new fusion scheme for SBP and DBP measurements was evaluated by the mean error (ME) and SDE between the estimated BP categories obtained from the classifier and target BP categories in accordance with the recommendations of the American National Standards Institute (ANSI)/American Association of Medical Instruments (AAMI) standard protocol [20], as shown in Table 3. BP devices are evaluated as passing on a standard protocol [20] basis if the ME is less than 5 mm Hg and the SDE is less than 8 mm Hg versus the reference.  Additionally, the British Hypertension Society (BHS) protocol determines the achievement of BP estimates with a cumulative percentage of readings belonging to the mean absolute errors of 5 mmHg, 10 mmHg and 15 mmHg in stethoscope measurements of two trained observers [28]. Moreover, it classifies the BP measuring device into four classes from A to D, as shown in Table 4.” (lines 165-174, page 10)

5. According to the comment “Data from patient are not clustered: for instance, Sex, BMI, Age (this last factor influence arterial stiffness and therefore the BP results) should be analyzed as factors affecting the outcomes.”

Answer 5. We analyzed and included the gender and age as our features in Figure 2.

6. According to the comment “No information about the computational time is provided: is the proposed solution to be ported into actual devices? if not why and what technological requirements should be improved?”

Answer 6. We provided about the total processing time such that “The total of the training and test times was calculated based on Matlab 2017 [27].  It was adequately short for real BP application because it processed 8,500 data in 6130.8 s. From this result, the DBM-based DS fusion called for a higher complexity time than the DBM technique, which was owing to the fusion process.

(lines 155-158, page 27)

Round 2

Reviewer 1 Report

The authors have addressed all my comments satisactorily and the paper can be considered for publication.

Reviewer 3 Report

The paper was improved according to the comments.